# Scratching the Surface Takes a Toll: Immune Recognition of Viral Proteins by Surface Toll-like Receptors

**DOI:** 10.3390/v15010052

**Published:** 2022-12-24

**Authors:** Alexis A. Hatton, Fermin E. Guerra

**Affiliations:** 1Department of Microbiology & Cell Biology, Montana State University, Bozeman, MT 59718, USA; 2Department of Laboratory Medicine & Pathology, University of Washington, Seattle, WA 98195, USA

**Keywords:** Toll-like receptors, TLR agonist, TLR antagonist, TLR2, TLR4, viruses, viral proteins, innate immunity, cytokines, interferons

## Abstract

Early innate viral recognition by the host is critical for the rapid response and subsequent clearance of an infection. Innate immune cells patrol sites of infection to detect and respond to invading microorganisms including viruses. Surface Toll-like receptors (TLRs) are a group of pattern recognition receptors (PRRs) that can be activated by viruses even before the host cell becomes infected. However, the early activation of surface TLRs by viruses can lead to viral clearance by the host or promote pathogenesis. Thus, a plethora of research has attempted to identify specific viral ligands that bind to surface TLRs and mediate progression of viral infection. Herein, we will discuss the past two decades of research that have identified specific viral proteins recognized by cell surface-associated TLRs, how these viral proteins and host surface TLR interactions affect the host inflammatory response and outcome of infection, and address why controversy remains regarding host surface TLR recognition of viral proteins.

## 1. Introduction

The innate immune system is a critical defender at host sites of entry against invading pathogens, granting the adaptive immune system the opportunity to develop an appropriate response to infection. Innate immune cells express a host of germline-encoded pattern recognition receptors (PRRs), including Toll-like Receptors (TLRs) that differentiate between bacteria, fungi, parasites, and viruses. The discovery of human TLRs began in 1996, when Bruno Lemaitre from Jules Hoffmann’s research group, linked *Drosophila* Toll to the activation of immune-related genes demonstrating differences between antifungal and antibacterial responses [1]. By 1997, a human homolog of *Drosophila* TLRs was discovered, which would later be named TLR4 [2]. In 1998, Bruce Beutler’s research team discovered that TLR4 was the PRR responsible for the recognition of lipopolysaccharide (LPS), a Gram-negative bacterial outer membrane component whose host recognition receptor had long eluded immunologists [3]. These discoveries would earn Beutler and Hoffmann the Nobel Prize in Physiology or Medicine in 2011 and ushered a pivotal era in immunology to determine how the innate immune system senses microbes.

Since 1998, the field of TLR immunology has rapidly expanded and identified that TLRs can detect microbe-specific ligands [1,2,3]. TLRs are primarily expressed in innate immune cells including macrophages, dendritic cells, neutrophils, and monocytes, but TLRs can also been found on non-innate immune cells including epithelial cells, fibroblasts, and adaptive immune cells [4,5]. To date, 10 human and 12 murine TLRs have been identified [4,6]. TLR1-9 are found in human and murine cells, while TLR10 is found in humans and TLR11-13 are found in mice [4,6]. TLRs bind components from bacteria, fungi, parasites, viruses, and host damage-associated molecular patterns (DAMPs). Although the TLR family recognizes a wide range of ligands, a cornerstone of TLR recognition is ligand specificity. TLR2/1 recognizes triacylated lipopeptides, TLR2/6 recognizes diacylated lipopeptides, TLR4 recognizes LPS, TLR5 recognizes flagellin, TLR3 recognizes double-stranded RNA (dsRNA), TLR7/8 recognizes single-stranded RNA (ssRNA), TLR9 recognizes CpG DNA, and TLR10’s ligand is not clearly established [6]. TLR ligand recognition is also spatially regulated. TLR1, TLR2, TLR4-6, and TLR10 encounter ligand on the cell surface [4,6]. TLR3, TLR7-9, and TLR11-13 bind ligand intracellularly in the endosomal compartment [4,6]. Here, we will focus on TLR recognition of viral proteins at the cell surface, which is one of the earliest events in viral pathogen-innate immune cell interaction and signaling. We refer the reader to the following reviews for intracellular TLR-ligand interactions and signaling [4,6].

## 2. Surface Toll-Like Receptor Signaling: Initiation

The majority of TLRs exist as homodimers, but TLR2 is unique due to its ability to heterodimerize. TLR2 has been shown to dimerize with several TLRs, including its primary dimers TLR1, and TLR6, but has been suggested to interact with TLR10 (Figure 1) [7,8]. TLR2 pathogen-associated molecular pattern (PAMP) recognition and signaling have also been shown to involve co-receptors such as CD14, and CD36 (Figure 1) [9]. Because of TLR2’s ability to heterodimerize with several other TLRs, the list of PAMPs suggested to associate with TLR2 are substantial, including proteins from Epstein–Barr virus (EBV) [10,11], cytomegalovirus (CMV) [12,13,14], respiratory syncytial virus (RSV) [15], measles [16], hepatitis C virus (HCV) [17,18,19,20,21,22], herpes simplex virus -1 and -2 (HSV) [23,24,25,26,27,28,29,30,31,32,33,34,35,36,37,38], coronaviruses [39,40,41,42], parvovirus [43,44], and norovirus [45].

TLR4 is primarily found as a homodimer. For TLR4-LPS binding, the canonical pathway of LPS presentation to TLR4 involves the LPS binding protein (LBP) solubilizing LPS aggregates [46]. Subsequently, LBP transfers a monomer of LPS to the CD14 co-receptor, which transfers the Lipid A portion of LPS to myeloid differentiation factor 2 (MD-2) [46]. Finally, the formation of the MD-2/TLR4 complex is required to initiate signaling from the cell surface (Figure 1) [46]. Even though the presentation of LPS to TLR4 requires several LPS specific steps, TLR4 also has an extensive PAMP list, including antagonistic viral proteins from hepatitis B virus (HBV) [47,48,49,50], and agonistic viral proteins from RSV [51], HCV [52], papillomaviruses [53,54], and Ebola [55].

All TLRs share several commonalities including leucine rich repeats, a transmembrane domain, and an intracellular toll-interleukin-1 receptor (TIR) domain [4,6,56]. After receptor-ligand interaction, the TIR domain recruits signaling protein adaptors including myeloid differentiation primary response 88 (MyD88); Toll-interleukin-1 receptor (TIR) domain-containing adaptor protein (TIRAP/Mal); TRIF-related adaptor molecule (TRAM); and TIR-domain-containing adapter-inducing interferon-β (TRIF); and one unique inhibitory signaling molecule, sterile α- and armadillo-motif-containing protein (SARM) [4,6,56,57]. TLR4 is unique due to its ability to signal via the MyD88-dependent pathway (Mal-MyD88) and the MyD88-independent pathway (TRAM-TRIF) (Figure 1) [4,6,46,56]. Recent studies suggest that TLR2 may also utilize pathways other than the Mal-MyD88 pathway for the activation of type I interferons (IFNs) (Figure 1) [58,59]. This signaling range by TLR2 and TLR4 gives these receptors the potential for additional diversity in response to various ligands. The remaining surface TLRs signal via MyD88 [4,6,56]. Upon ligand binding, the TIR domain of the ligand bound receptor activates recruitment of its cognate signaling adaptor proteins resulting in the activation of the nuclear factor-κB (NF-κB) pathway, mitogen-activated protein kinase (MAPK) pathway, and/or interferon regulatory factors (IRF) pathway [4,6,56] (Figure 1, detailed TLR signaling reviews are included here [4,6]).

## 3. Surface TLR Associated Virus Interactions

We will focus on the vast body of literature dedicated to determining how surface TLRs recognize viruses. Studies identifying viral proteins that interact with surface TLRs typically focus on viral proteins present on the virus surface, but we will also highlight non-structural proteins and proteins secreted during infection that interact with surface TLRs. Studies reviewed here will analyze both infectious viruses and replication-deficient viruses to demonstrate that active replication does not always lead to surface TLR recognition. In addition, we will examine studies that utilized various methods to determine viral ligand and surface TLR-dependent signaling including the use of purified viral proteins, gene deletion, and viral protein-neutralizing antibodies. While TLR2 bacterial and fungal ligands have been described in detail elsewhere [60,61], herein we will focus on viral ligands identified as associating with surface TLRs and potential co-receptor involvement. For a table of viruses recognized by TLR2, TLR1, TLR6, TLR4, TLR5, and TLR10, refer to Appendix A. The viruses discussed here are subdivided by their Baltimore Classification [62,63].

### 3.1. Baltimore Classification: Class I

#### 3.1.1. Cytomegalovirus

Cytomegalovirus (CMV) is mainly a complication for newborns, pregnant women, and the immunocompromised causing mononucleosis, end organ disease, and nephritis [64]. CMV is an enveloped, double-stranded DNA virus in the *Herpesviridae* family. The CMV host-derived envelope contains 3 primary virus-derived glycoprotein complexes: I, II, and III [65]. These complexes contain varying compositions of the 6 glycoproteins including gB, gH, gL, gM, gN, and gO depending on the complex [65]. CMV has also been demonstrated to stimulate the secretion of defective viral particles containing only glycoproteins and tegument proteins which can induce apoptosis and antibody production against gB and gH [65]. One study identified 10 viral proteins present in the secretome during human CMV (HCMV) infection, of which only 4 were known proteins including UL32 (pp150), UL44, UL122, and UL123 [66]. This suggests that proteins other than surface viral proteins could be present for the activation of surface TLRs during CMV infection. Although a vaccine does not yet exist for CMV, two groups demonstrated that a pentameric glycoprotein complex consisting of gH, gL, UL128, 130, and 131 was a better candidate for inducing a neutralizing antibody response than gB alone [67]. While vaccines are still in development, understanding how exogenously expressed proteins activate an immune response is critical for vaccine development and understanding the complexity of host–virus interactions.

Compton et al. first demonstrated that TLR2 interacted with CD14 for the recognition of HCMV virions and induced the production of inflammatory cytokines including interleukin 6 (IL-6) and IL-8 [12]. In this study, infectious, ultraviolet inactivated (UV-IA), and HCMV defective particles all induced IL-8 protein production in HEK-293 cells expressing TLR2/CD14 but not TLR4/CD14 [12]. In support, Boehme et al. further demonstrated that HCMV recognition was dependent upon TLR2 in HEK-293 cells and human fibroblasts [13]. The TLR2/1 heterodimer, but not TLR2/6, was found to interact with the glycoproteins gB and gH, but not gL, through co-immunoprecipitation [13]. Using UV-IA HCMV and antibodies against gB and gH, it was confirmed that both antibodies diminished the production of IL-6, suggesting that specific HCMV proteins were responsible for the activation of TLR2/1 [13]. A study by Brown et al., utilized purified HCMV gB to demonstrate that HEK-293 cells expressing TLR2 responded to gB, resulting in a modest increase in *IL6* and interleukin 1 beta (*IL1B*) expression, and a marked increase in IL-8 and tumor necrosis factor (TNF) production [14]. Importantly, this group previously demonstrated that the TLR2 single nucleotide polymorphism (SNP) R753Q was associated with a higher degree of HCMV replication and liver disease [68] and then demonstrated in vitro in HEK-293 cells that this TLR2 SNP impaired the activation of NF-κB and subsequent regulation of these cytokines in response to gB [14]. Thus, these studies suggest that TLR2 by formation of the TLR2/1 heterodimer interacts with the HCMV glycoproteins gB and gH to mount an immune response (Figure 2).

#### 3.1.2. Epstein–Barr Virus

Epstein–Barr virus (EBV) is estimated to infect >95% of the world population by adulthood and is the cause of ~125,000 cases of infectious mononucleosis annually [69]. EBV is also one of the first known oncogenic viruses and is associated with several lymphoid and epithelial cancers [69]. The typical route of transmission for EBV is via saliva where EBV then undergoes lytic or latent infection [69]. EBV is a double-stranded DNA virus in the *Herpesviridae* family with an envelope consisting of 13 potential glycoproteins depending on the stage of infection [69,70]. Additionally, studies have indicated that EBV deoxyuridine triphosphate nucleotidohydrolase (dUTPase), expressed during the lytic cycle of EBV, induces a sickness response in mice suggesting that several EBV-associated proteins could interact with surface TLRs resulting in host immune activation [10,11,71].

Gaudreault et al. first demonstrated that UV-IA and replication inhibited EBV activated a TLR2-dependent NF-κB response in HEK-293 cells [72]. Research by this group further identified a potential EBV glycoprotein responsible by utilizing monoclonal antibodies against EBV gp350/220 [72]. The gp350/220 antibody partially prevented the TLR2 induced response to EBV, but as it did not completely abrogate the response to EBV, it was suggested that the antibody may instead sterically hinder the actual TLR2-EBV ligand [72]. While Gaudreault et al. suggested that EBV gH or gB may be alternative TLR2 activating ligands, Ariza et al. instead identified dUTPase as the potential TLR2 ligand [10,72]. Ariza et al. focused on lytic-related EBV proteins including dUTPase and demonstrated that dUTPase activated human TLR2 but not TLR4 in HEK-293 cells, resulting in the production of IL-6 [10]. Interestingly, this group demonstrated that neither TLR1 nor TLR6 overexpression in HEK-293 cells enhanced the response to EBV dUTPase but did suggest that TLR6 may serve as a negative regulator of the TLR2-dUTPase response [10]. Ariza et al. later demonstrated that EBV dUTPase present in exosomes activated the production of proinflammatory T_H_1/T_H_17 cytokines in a TLR2-dependent manner [11]. Two additional studies further demonstrated that EBV induced a TLR2-dependent response resulting in production of inflammatory cytokines/chemokines including IL-8 and monocyte chemoattractant protein 1 (MCP-1) [73,74] and the anti-inflammatory cytokine, IL-10 [73] in primary human monocytes [73] and THP-1-differentiated macrophages [74] potentially contributing to tumorigenesis associated with EBV infection. Therefore, several studies demonstrated a role for host TLR2 and dUTPase, gB, gH, and gp350/220 (Figure 2) in the recognition of EBV, but questions remain whether the response by TLR2 benefits the host or virus and remains an area of active research [11,74]. Furthermore, the immunomodulatory role of TLR6 in response to EBV remains to be fully elucidated.

#### 3.1.3. Herpes Simplex Virus

Globally, millions of people are exposed to herpes simplex virus (HSV), which is primarily attributed to two α-herpesvirus serotypes including HSV-1 and HSV-2 [75]. The prevalence of HSV-1 and HSV-2 worldwide was estimated to be 66.6% in 0–49-year old’s and 13.2% in 15-49-year old’s [76]. While HSV is typically associated with mild disease including cold sores, HSV can cause more severe pathologies including encephalitis [75]. HSV establishes lifelong, latent infections in the host’s peripheral nervous system and can be reactivated by periods of stress [77,78]. HSV are enveloped, double-stranded DNA viruses in the *Herpesviridae* family. The envelope of HSV-1/-2 contains approximately 12 different glycoproteins, of which, 4 are necessary for entry including gH, gD, gL, and gB [75,79]. In addition to these glycoproteins, there are non-glycosylated proteins that can be present in the lipid envelope including UL20, UL45, and US9 [79].

HSV-1 was first suggested to contain a TLR2 ligand (s) by Kurt-Jones et al. and then further suggested by Aravalli et al. [23,29]. Both studies demonstrated in mice that the recognition of HSV-1 resulted in detrimental inflammation including MCP-1 [23], IL-6 [23,29], TNF, IL-1β, and other chemokines [29], while MCP-1 and IL-6 were also produced in response to UV-IA HSV-1 suggesting this was a replication-independent recognition mechanism [23]. The absence of TLR2 but not TLR4 increased the survival of mice after HSV-1 infection as none of the mice experienced paralysis or seizures [23]. Subsequently, Kurt-Jones et al. further suggested that TLR2 recognition of HSV-1 and HSV-2 is detrimental in neonates leading to increased production of IL-6 and IL-8 by neonatal blood cells, which could explain why sepsis syndrome is observed more frequently in neonates with HSV [24]. Studies on HSV-induced apoptosis suggest that upregulation of inflammatory cytokines and pro-apoptotic genes depends on TLR2 responding to HSV, including the upregulation of IL-15 [32], and the absence of TLR2 signaling reduced cell death during HSV-1 infection [31]. More recently, Brun et al. linked the C-C motif chemokine ligand 2 (CCL2)-dependent recruitment of macrophages to TLR2-HSV-1 recognition [80]. The absence of TLR2 in C57BL/6 mice resulted in increased transcription of *Cxcl11* and *Cxcl9* suggesting that the absence of TLR2 skewed the immune response from recruiting macrophages to T cells [80]. However, other studies have identified contradicting results. Mansur et al. showed that while TLR2 contributed in part to an inflammatory response during HSV-1 infection in mice and CHO cells, TLR2 and TLR4 alone were not sufficient to respond to HSV [81]. Liu et al. demonstrated that TLR4 was involved in the production of inflammatory IL-6 and the antiviral type I IFN interferon-β (IFN-β) in human cervical epithelial cells, but TLR2 was not tested [82,83]. Following these studies, Lv et al. also demonstrated the involvement of TLR4 in HSV-2 recognition in human genital epithelial cell lines, while TLR2 and TLR9 were upregulated in response to HSV-2 [84].

To identify the viral ligands recognized by the host, Leoni et al. utilized HSV deficient in either gD, gB, or gH or utilized purified gD, gH, gL, and gB proteins to demonstrate that while gH/gL, and gB bind to TLR2, only gH/gL induce a signaling response via NF-κB in HEK-293 cells [34]. In contrast, Cai et al. demonstrated that gB was responsible for activating a TLR2-dependent NF-κB response and that TLR4 was not involved in gB recognition in HEK-293 cells [35]. Several other studies have suggested the involvement of gH, gL and/or gB as TLR2 ligands [38,80,85]. While Leoni et al. and Reske et al. analyzed the inflammatory response to multiple HSV glycoproteins [34,86], other studies have only focused on single glycoproteins [35,87]. Ariza et al. instead indicated the involvement of the non-structural herpesvirus encoded dUTPase [36]. Ariza et al. identified that the dUTPase from several different herpesviruses including HSV-2, human herpesvirus-6A, human herpesvirus-8, and varicella-zoster virus, as well as EBV [11], all induced NF-κB signaling and the secretion of proinflammatory cytokines via TLR2/1 and partially via TLR6 in HEK-293 cells [36]. In addition, HSV-associated TLR2, and potentially TLR4, antagonistic proteins have also been identified including ICP0 [88,89] and Us3 [90]. Thus, the TLR2 HSV ligand(s) remains under debate. These studies, while not identifying a definitive TLR2 agonist or antagonist, demonstrate that HSV, along with other viruses in the *Herpesviridae* family, likely contain TLR2 ligands.

One possible explanation for the discrepancy between studies is the variety of HSV laboratory strains and clinical isolates analyzed. A study by Sato et al. attempted to address the variability found between HSV laboratory strains and clinical isolates by utilizing different HSV strains [30]. This study concluded that only a rare population of strains, including HSV-1 KOS strain (HSV-K/KOS) and HSV WT 186 used by Kurt-Jones et al., and unique long region 29-deficient 186 strains (186-K and UL29-186-K), activate TLR2 in HEK-293 cells [23,24,30]. Sato et al. also identified a compensatory role for TLR9 in addition to TLR2 in the recognition and response to HSV [30]. Several other studies further investigated the role of TLR2 in combination with TLR9 in responding to HSV and whether this response benefitted the host or virus [26,28,33,38,91]. These studies demonstrate that the combination of TLR2 and TLR9 provide protection from mortality and encephalitis in mice [33,37,38,91], but studies by Sarangi et al. and Guo et al. instead demonstrate that these receptors in mice contribute to the inflammatory disease associated with HSV infection [26,28]. Additionally, Sarangi et al. demonstrated that TLR4 may have a role in mediating an anti-inflammatory response as TLR4-deficient mice experienced more severe lesions than wild-type mice [26]. In contrast to these studies, Wang et al. demonstrated that while TLR2 was involved in detrimental inflammatory cytokine production after intracranial infection, TLR9 was not involved in HSV-associated mortality, IFN-α/β production, or viral burden in mice [27]. Future studies should resolve these discordant results to determine the surface TLR (s) involved in HSV recognition, the HSV ligand (s) involved in host recognition, and whether recognition benefits the host or virus (Figure 2).

#### 3.1.4. Human Papillomavirus

Human papillomavirus (HPV) is associated with 90% of cervical cancers diagnosed and can cause genital warts [92]. HPV is a double-stranded DNA virus from the *Papillomaviridae* family that is non-enveloped and consists of two capsid proteins including the major capsid protein, L1, and minor capsid protein L2 [93]. To date, there are limited studies on HPV signaling. However, two studies have suggested that virus-like particles (VLPs) composed of the HPV capsid protein L1 are recognized by surface TLR4 in human monocyte-derived dendritic cells and in mice [53,54] (Figure 2). Yang et al. suggested that VLP-based HPV vaccines may provide the host with a T helper cell-independent humoral response benefitting CD4^+^-deficient patients that are prone to HPV disease [54].

#### 3.1.5. Varicella-Zoster Virus

Varicella-zoster virus (VZV) is the causative agent of chickenpox and upon reactivation years to decades later, causes shingles [94]. VZV is a double-stranded DNA virus from the *Herpesviridae* family. The envelope of VZV contains 8 glycoproteins including gB, gH, and gL that form the core fusion complex, and gC, gE, gI, gK, and gN that may contribute as accessory proteins [95]. While glycoproteins from VZV have not been identified as surface TLR agonists, Ariza et al. identified that VZV dUTPase activated an inflammatory response via TLR2/1 in HEK-293 cells [36] (Figure 2). Wang et al. identified a role for TLR2 in the recognition of VZV, but only human cells were able to respond [96]. Interestingly, murine RAW264.7 cells expressing human TLR2 could not respond to VZV, but HEK-293 cells expressing murine TLR2 could respond to VZV [96]. This observation suggests human or murine TLR2 can respond to VZV if the co-factors upstream of TLR2 are of human origin [96]. In contrast, Yu et al. demonstrated that TLR9, not TLR2 in human peripheral blood mononuclear cells (PBMCs), was involved in the recognition of VZV [97]. However, Wang et al. analyzed a proinflammatory response while Yu et al. focused on IFN-α, which may explain the observed differences in TLR requirement for VZV recognition [96,97].

### 3.2. Baltimore Classification: Class II

#### Parvovirus

While porcine parvovirus (PPV) is not a human or murine virus as we have discussed thus far, it has been identified as an activator of TLR2 and is worth mentioning in the case that a role for TLR2 is identified for the human parvovirus, B19 [98]. PPV can be found in most pig herds throughout the world and is a major cause of porcine reproductive failure [99]. PPV is a non-enveloped, single-stranded DNA virus in the *Parvoviridae* family and encodes two non-structural proteins, NS1 and NS2, and two structural proteins, VP1 and VP2 [99]. While little research has been done to investigate how PPV activates a surface TLR, there have been two studies demonstrating that the recognition of PPV by TLR2 may benefit the virus [43,44]. Jin et al. demonstrated that PPV NS1 expressed by the pEGFP-N1-NS1 plasmid in HEK-293 cells induced the expression and production of TNF and IL-6 and that NS1-NF-κB signaling activation was mediated by TLR2 [43]. Xu et al. used porcine PK-15 cells to demonstrate that infectious PPV induced apoptosis in a TLR2-dependent manner [44]. Interestingly, infectious PPV was required to activate NF-κB signaling as UV-IA PPV did not [44]. These studies suggested that PPV may utilize TLR2 to modulate the host response including inducing cytokine storm during PPV infection [43,44]. As UV-IA PPV did not induce a response and NS1 was required to activate TLR2, these studies suggest that surface TLRs may be activated in response to PPV post-replication (Figure 3).

### 3.3. Baltimore Classification: Class III

Class III viruses include the family of *Reoviridae*, which are double-stranded RNA viruses [100]. One virus in the *Reoviridae* family known to infect humans is rotavirus causing acute gastroenteritis in children [101]. To our knowledge, studies with Class III viruses activating surface TLRs have not been widely identified. However, a study by Ge et al. identified that purified rotavirus non-structural protein 4 (NSP4) induced IL-8 production in TLR2-expressing HEK-293 cells but not TLR4-expressing cells [101]. In vivo studies to determine the role of surface TLRs in the recognition of rotavirus are complicated by NSP4, an enterotoxin, which can induce gastroenteric damage and may indirectly induce inflammatory response through host DAMPs or PAMPs from the disrupted microbiota [102,103].

### 3.4. Baltimore Classification: Class IV

#### 3.4.1. Dengue Virus

Dengue viruses (DENV) are made up of four serotypes that result in a variety of clinical manifestations including vomiting, leukopenia, liver enlargement, and severe bleeding [104]. Individuals who experience secondary dengue infections are at risk for antibody-dependent enhancement resulting in increased disease severity [104]. DENV is an enveloped virus consisting of a single-stranded, positive sense RNA genome in the *Flaviviridae* family primarily transmitted by the *Aedes aegypti* mosquito vector [104]. While there is a commercially available vaccine against DENV, Dengvaxia^®^, there is ongoing research into the development of more effective vaccines for children under the age of 9 [105]. DENV encodes three structural proteins including membrane (M), envelope (E), and capsid (C) protein [104]. As we will discuss, a few studies have also indicated a role for DENV NS1 in activating an innate response as this protein can be secreted as soluble NS1 during infection [104,106,107,108].

Chen et al. first demonstrated a potential involvement for surface TLRs by showing that TLR6 was upregulated in the K562 lymphoblast cell line in response to DENV infection [109]. In a subsequent study, Chen et al. provided a detailed analysis of the TLR responsible and the DENV protein potentially involved in this response [106]. In agreement with the first study, they found that TLR6, as well as TLR2 were upregulated in human PBMCs in response to DENV resulting in the production of TNF and IL-6 [106]. Importantly, NS1 was responsible for the changes in TNF and IL-6 production in a TLR2- and TLR6-dependent mechanism [106]. Finally, Chen et al. demonstrated that TLR6-deficient mice displayed increased survival after DENV infection compared to wild-type mice, suggesting that TLR6 has a detrimental role in DENV pathogenesis [106]. In validation of NS1 playing a detrimental role in inflammation during DENV infection, two additional studies also identified NS1 as activating surface TLRs to produce inflammatory cytokines [107,108]. These studies demonstrated that NS1 played a role in vascular leakage and activating inflammatory cytokine expression and/or production including TNF and IL-6 but via TLR4 and not TLR2 in murine bone marrow-derived macrophages and human PBMCs [107,108]. Modhiran et al. attribute previous studies observing a role for TLR2/6 potentially as the result of *Escherichia coli* contaminants present in the NS1 preparation among other variables [108]. Two additional studies by Aguilar-Briseño et al. explored the role of surface TLRs in responding to DENV but did not identify specific proteins [110,111]. These studies examined all DENV serotypes, identifying that all serotypes activate TLR2/6, not TLR1 or TLR4, resulting in the expression and production of canonical inflammatory cytokines including TNF and IL-6 in human PBMCs [110,111]. In addition, immature DENV particles, which are non-infectious in the absence of anti-DENV antibody, activated TLR2/6 in human PBMCs suggesting that the innate immune system senses DENV independent of viral replication [111]. This suggests that in addition to NS1, which is produced post-infection, DENV contains ligands present in the envelope that activate TLR2 (Figure 4).

#### 3.4.2. Hepatitis C Virus

Hepatitis C virus (HCV) can cause chronic infection resulting in liver damage including cirrhosis and hepatocellular carcinoma and does not currently have a vaccine [112]. HCV is an enveloped, positive-sense RNA virus in the *Flaviviridae* family. HCV “spikes” protrude from the envelope and are composed of E1 and E2 heterodimeric glycoproteins [113]. The outside of HCV is also able to interact with host-derived lipoproteins termed lipoviral particles [113]. While the host-derived lipoproteins associated with HCV vary, HCV most frequently associates with low density lipoproteins, very low-density lipoproteins, and apolipoproteins (Apo) A1, B, C and E [113]. The HCV core protein and non-structural protein 3 (NS3) are not present on the viral envelope but can be found outside the host cell during infection and can be found in the serum of HCV patients [17,114]. In addition to proteins exposed on the viral envelope, HCV produces defective interfering RNAs capable of infection and replication but cannot be packaged into viral particles without the presence of compensatory wild-type HCV [115]. HCV particles carrying these defective genomes are suggested to function as “immunological decoys” [115].

Early HCV surface TLR recognition studies focused on HCV core protein and NS3, which activated TLR2, but not TLR4, and signaling was MyD88-dependent [17,20,21,22]. While TLR2 was not necessary for the internalization of these proteins, TLR2 activation was necessary for the production of TNF, IL-6, and IL-8 in different mammalian cells [17,20,21,22]. Both Chung et al. and Swaminathan et al. suggest that HCV core protein via TLR2 signaling may increase host susceptibility to microbial infection [20] and HIV infection [21]. Dolganiuc et al. additionally examined NS4, NS5 and E2, which failed to produce an inflammatory response or induced a weak response compared to HCV core protein and NS3 in human monocytes and HEK-293 cells [17]. Finally, they identified that the portion of NS3 necessary for TLR2 interaction was within the amino acid sequence, 1450–1643 [17]. Chang et al. expanded on their TLR2 heterodimer studies to demonstrate that surface TLR recognition of HCV proteins is complex [18]. Primary human monocyte-derived macrophages produced TNF in response to NS3 through TLR6 but depended on TLR2/1 or TLR2/6 to recognize HCV core protein and produce TNF [18]. The cytokine response also depended on specific TLR recognition of HCV proteins [18]. The production of IL-6 by human monocyte-derived macrophages depended on TLR2/1 or TLR2/6 in response to HCV core protein and NS3, but the production of IL-10 depended on TLR2/1 for NS3 recognition and only TLR2 for HCV core protein [18]. These results did not fully align with their previous publication as TLR1 did not have a role in human monocytes [17]. Interestingly, they found that while TLR2/6 was involved in mice during infection with HCV, TLR1 was not involved indicating differences between human and mouse TLR2/1 or TLR2/6 in the HCV response [18]. Machida et al. examined the role of similar HCV proteins, including core, E1, E2, NS3, NS4B, NS5A, and NS5B, and only NS5A resulted in TLR4 activation in hepatocyte and B cell cell-lines but TLR1, TLR2, and TLR6 were not examined [52]. Additionally, this group demonstrated that the production of inflammatory IL-6 and the type I IFN, IFN-β, in response to HCV were TLR4-dependent [52]. Zhang et al. demonstrated a unique role for HCV core, whereby HCV core protein prevented human monocyte-derived macrophage polarization to M1 or M2 via TLR2 and STAT1 or TLR2 and STAT3 signaling, respectively [116]. Defective macrophage polarization in HCV-infected individuals may contribute to HCV persistence in patients [116]. These studies suggest that the TLR2 heterodimeric response to HCV proteins is complex, and cytokine production is dependent on specific HCV protein-TLR interactions (Figure 4).

The relevance of these publications was questioned by Hoffmann et al. [19]. This group compared purified HCV core protein to either infectious HCV or HCV VLPs and found that while purified HCV core protein triggered TLR2-dependent signaling, infectious HCV, HCV VLPs, or purified E1 or E2 did not [19]. Although HCV-infected Huh7.5 cells released a comparable level of core protein as seen in HCV-infected patients, a role for TLR2 was not observed [19]. These results suggest that while TLR2 may recognize purified HCV core protein, TLR2 is dispensable during infection [19]. The comparison of fully infectious virus to specific purified protein is important to demonstrate biological relevance of surface TLR-viral protein interaction. It is also important to recognize that virus–host interactions are complex and observing a response to infectious virus in the absence of a single receptor does not negate the role of that receptor, as there are likely compensatory mechanisms employed by the host.

#### 3.4.3. Severe Acute Respiratory Syndrome Coronavirus

The severe acute respiratory syndrome coronavirus (SARS-CoV) family causes respiratory infections that can lead to asymptomatic infection, shortness of breath, pneumonia, cardiac injury, and multiple organ failure [117]. The current global pandemic caused by SARS-CoV-2 has infected over 600 million individuals, resulting in more than 6.5 million deaths [118]. The *Coronaviridae* family of viruses are enveloped, with a positive sense, single-stranded RNA genome. The envelope consists of four primary structural proteins including spike (S), nucleocapsid (N/NC), envelope (E), and membrane (M) proteins [119]. S protein forms a trimer that mediates entry into the host cell [119]. The structure of S protein consists of the S1 region containing the receptor binding domain (RBD) that is surface exposed and the transmembrane S2 domain that contains the fusion peptide [119].

Previous studies have analyzed surface TLR activation in response to SARS-CoV. In 2009, Dosch et al. published a study demonstrating a role for SARS-CoV-1 S protein in the activation of TLR2 [120]. Using HEK-293 cells expressing human TLR2, they demonstrated that S protein activated IL-8 production in a TLR2-dependent manner, and that E protein did not [120]. With the SARS-CoV-2 pandemic, many studies quickly began investigating the role of structural proteins in the activation of surface TLRs and how this activation may contribute to the initiation of cytokine storm associated with infection. As studies into SARS-CoV-2 are ongoing and numerous studies have been published since the initial outbreak, this section will focus on select, peer-reviewed surface TLR-related studies.

The role of SARS-CoV envelope proteins in activating surface TLRs remains under debate. Sohn et al. first identified that TLR4, CD14, and additional receptors were upregulated in SARS-CoV-2 infected individuals [121]. While they did not directly demonstrate a role for TLR4 in the recognition of SARS-CoV-2 structural proteins, they did demonstrate that S2 and N protein upregulated the transcription of *IL6*, *TNF*, *IL1B*, and *IL12p40* [121]. Interestingly, while S1 and the RBD did not upregulate the expression of proinflammatory cytokines, they did express the type I IFN, *IFNB*, but S2 did not [121]. In contrast, Frank et al. demonstrated that S1, not S2, induced an inflammatory response in an in vivo rat model and in HEK-293 cells expressing human TLR2 or TLR4, but a more robust response in cells expressing TLR2 [39]. However, two studies suggested that the S1 protein was involved in the activation of an inflammatory response in a TLR4-dependent mechanism and that TLR2 was not involved in murine microglial cells, peritoneal macrophages, RAW264.7, and human THP-1-differentiated macrophages [122,123]. Several studies have demonstrated that E protein, not S protein, is recognized by TLR2 in HEK-293 cells, peritoneal dendritic cells, murine bone marrow-derived macrophages, and PBMCs [40,41,42]. Other studies have demonstrated that S protein is indeed responsible for activating an inflammatory response, but that the full S protein trimer is involved in activating an inflammatory response [124,125]. These studies disagree on which surface TLR mediates activation by the S protein trimer. Khan et al. showed that TLR2, with TLR1 and TLR6 acting in a compensatory manner in HEK-293 and RAW264.7 cells, were required for the response to the S protein trimer [125], whereas Zhao et al. suggested that TLR4 was necessary in THP-1 monocytes, murine peritoneal macrophages, RAW264.7, and bone marrow-derived macrophages [124]. Thus, there is no consensus on which SARS-CoV-2 protein(s) are responsible for the activation of surface TLRs, nor is there consensus on which TLR is responsible (Figure 4). The lack of consensus could be in part due to the variety of cell types used in these studies. While a consensus has not been made on which SARS-CoV-2 proteins activate which surface TLRs, several studies suggest that SARS-CoV-2 proteins contribute to detrimental cytokine storm via TLR2 [40,125] or TLR4 [121,123] associated with severe disease during infection. In addition, studies have demonstrated that detrimental neuroinflammation and subsequent central nervous system pathology is associated with the activation of TLR2 [39], and/or TLR4 [39,122].

#### 3.4.4. Yellow Fever Virus

Yellow fever virus (YFV) is transmitted by the *Aeges aegytpi* mosquito causing Yellow Fever and while vaccination is available, the virus is continuing to spread to areas of the world where vaccination is not prevalent [126]. YFV is an enveloped virus with a positive-sense, single-stranded RNA genome. Three structural proteins are encoded by YFV including capsid (C), membrane (prM/M), and envelope (E) along with 7 non-structural proteins [126]. Modhiran et al. identified that NS1 from several *Flaviviridae*, including YFV, induced the secretion of IL-6 and that this response was likely mediated by TLR4 in PBMCs (Figure 4) [107]. Querec et al. focused on how the live-attenuated YFV vaccine, YF-17D, activated an immune response [127]. They demonstrated that TLR2 played a role in the production of IL-12p40 and IL-6 and that TLR2 may regulate the Th1/Th2 balance in response to YF-17D in mice and murine and human dendritic cells [127]. Additionally, UV-IA YF-17D altered the cytokine response, but IL-12p40 was still produced in the absence of viral replication [127].

### 3.5. Baltimore Classification: Class V

#### 3.5.1. Ebola Virus

Ebola virus (EBOV) is the causative agent of the 2013 to 2016 epidemic in Africa, causing over 11,000 reported deaths [128]. EBOV outbreaks are usually traced to a single spillover event and transmitted by direct contact [129]. Disease associated with EBOV can consist of renal dysfunction, cytokine-mediated nephrotoxicity, respiratory symptoms including pulmonary oedema, and hemorrhagic manifestations [129]. EBOV is a single-stranded, negative sense RNA virus in the *Filoviridae* family. The RNA genome of EBOV encodes 7 proteins with only glycoprotein (GP) being surface exposed [130]. GP can either be a component of the viral membrane, shed, or secreted as a soluble, non-structural GP (nsGP) [131,132]. VP40 is the viral matrix protein that binds the viral envelope and alone can form VLPs in mammalian cells [130].

Several studies have indicated a role for TLR4 in the recognition of EBOV and that TLR4-EBOV interaction is detrimental to the host possibly by increasing vascular permeability and increased susceptibility to EBOV infection due to increased monocyte differentiation [55,133,134,135,136,137]. Using VP40 or VP40 + GP VLPs, Okumura et al. demonstrated that VP40 VLPs elicited minimal inflammatory cytokine production, but the addition of GP to the VLPs induced a strong inflammatory response in both THP-1 monocytes and HEK-293 cells [55]. GP-mediated activation of an inflammatory response via NF-κB signaling was TLR4-dependent [55]. Escudero-Pérez et al. demonstrated a role for shed GP in the activation of TLR4, whereas nsGP could bind TLR4 without activating human dendritic cells and macrophages [133]. Iampietro et al. demonstrated in two studies how exposure to shed GP increased human monocyte and T cell differentiation and increased cell death by apoptosis and necrosis via TLR4 (Figure 5) [134,137]. In terms of treatments against EBOV infection, Younan et al. demonstrated that eritoran, a TLR4 antagonist, could serve as a therapeutic strategy and increased mouse survival following EBOV infection by reducing viral burden [136]. In addition to therapeutics, Lai et al. suggested that GP may stimulate protective immunity through TLR4 activating an innate-adaptive immune response, which could benefit vaccine development [135].

#### 3.5.2. Lymphocytic Choriomeningitis

During the mid-1900’s, lymphocytic choriomeningitis (LCMV) was identified as one of the leading causes of aseptic meningitis in humans. LCMV can also cause mortality in individuals following organ transplant [138]. The host of LCMV is the common house mouse, and while infections are thought to be underreported in humans, it is still believed to be an important cause of meningitis in humans [138]. LCMV is an enveloped virus in the *Arenaviridae* family consisting of a negative sense, singled-stranded RNA genome. The envelope of LCMV contains a glycoprotein precursor, which upon proteolytic processing generates 3 subunits consisting of the stable signal peptide, glycoprotein 1, and glycoprotein 2 [139].

Zhou et al. identified that LCMV activated a TLR2-dependent response to produce IL-6 or IL-8 in mouse peritoneal macrophages or HEK-293 cells, respectively (Figure 5) [140]. UV-IA LCMV induced IL-8 production in HEK-293 cells but to a lower extent compared to infectious LCMV, suggesting that replication of this virus was not necessary for TLR2 activation [140]. Zhou et al. demonstrated that TLR2-deficient mice did not have altered viral clearance but were unable to produce MCP-1 and IFN-α [140]. A subsequent study demonstrated that the LCMV-TLR2 interaction was also observed with mouse-derived astrocytes and microglial cells [141]. Interestingly, peritoneal macrophages, astrocytes and microglial cells displayed different cytokine expression patterns in response to LCMV [141]. Astrocytes primarily produced MCP-1 and microglial cells primarily produced TNF in response to LCMV, whereas mixed glial cells did not produce any IL-6 as was observed previously with peritoneal macrophages [140,141]. Zhou et al. suggested that an exacerbated inflammatory response resulting in increased inflammatory cell recruitment to the brain may increase the risk of damage to the central nervous system during infection [141]. These studies suggest that cytokine responses to LCMV are cell-type dependent but reliant on TLR2 [141].

#### 3.5.3. Measles

Measles virus causes a highly contagious respiratory infection commonly associated with severe pathogenicity in children but is preventable by vaccination [142,143]. Measles virus is an enveloped, single-stranded RNA virus in the *Paramyxovirus* family. The measles virus host-derived envelope contains 2 attachment glycoproteins, hemagglutinin (HA) and fusion glycoprotein (F) [142]. Nucleocapsid (NC) protein is the most abundant measles virus protein and has been suggested to interfere with the immune response via binding the Fc portion of immunoglobulin G, but how nucleocapsid protein is secreted from the cytoplasm to readily induce this inhibition is not well understood [144]. The live attenuated measles vaccine is primarily derived from the Edmonston strain, which is attenuated through passaging [143].

Vaccine strain measles viruses display differences from wild-type measles virus in most viral proteins produced, but one highlighted difference is the asparagine to tyrosine amino acid change at position 481 of the HA protein [16,143]. Bieback et al. investigated TLR2-dependent differences in immunogenicity between wild-type measles virus and vaccine strains of measles virus, particularly in relation to the asparagine to tyrosine mutation in measles virus HA protein [16]. Using CHO cells stably expressing human CD14 along with human TLR2 or human TLR4 and an NF-kB-dependent CD25 reporter gene, the wild-type measles virus strain activated the CD25 reporter only in CHO cells expressing hCD14/TLR2, whereas the vaccine strain of measles virus did not activate the reporter [16]. Using recombinant wild-type and vaccine strain measles viruses, it was demonstrated that allelic exchange of the HA protein in the vaccine strain for the wild-type HA stimulated the reporter, whereas swapping the fusion protein for the wild-type fusion protein did not, suggesting that wild-type measles HA protein alone is sufficient to activate TLR2 [16]. Wild-type measles HA additionally induced transcription of *IL1A*, *IL1B*, *IL12B*, and *IL6*, whereas vaccine strain HA did not [16]. To further confirm that TLR2 and CD14 were involved in the recognition of wild-type measles HA, anti-TLR2 and anti-CD14 antibodies were used to partially inhibit *IL6* expression [16]. These results clearly suggest an involvement for measles HA in surface TLR2 recognition (Figure 5).

#### 3.5.4. Respiratory Syncytial Virus

Respiratory syncytial virus (RSV) causes severe lower respiratory tract disease in infants and children, and an RSV vaccine has not been developed [145]. RSV is an enveloped, single-stranded RNA virus in the *Paramyxovirus* family. The RSV host-derived envelope contains 3 membrane-associated proteins including a fusion protein (F), a short hydrophobic protein, and an attachment glycoprotein (GP), the latter can also be secreted during infection presenting several candidate proteins for surface TLR interaction [146,147]. RSV was the first virus to be directly associated with a surface TLR, and while the first publication by Kurt-Jones et al. focused on TLR4, they also discussed the potential involvement of TLR2 [51].

Kurt-Jones et al. primarily focused on RSV-TLR4 interaction, specifically comparing whole virus and individual proteins including GP, F, and nucleocapsid (NC) protein. Several inflammatory cytokines including IL-6, IL-8, and TNF depended on TLR4-RSV F protein recognition in purified human monocytes [51]. TLR4 recognition was critical for the control of RSV infection [51]. Regarding TLR2, Kurt-Jones et al. briefly discussed that TLR2 transfection in CHO cells enhanced NF-κB activation in response to RSV F protein, the first hint that TLR2 may have a role in the response to RSV [51]. Several studies also established a role for TLR4 during RSV infection, including Tulic et al. that demonstrated that the TLR4 mutations Asp299Gly and Thr399Ile were associated with severe RSV bronchitis in infants [148,149,150,151]. Furthermore, and in support, Funchal et al. demonstrated that F protein activated a TLR4-dependent response in human neutrophils causing the release of neutrophil extracellular traps [152]. Indeed, RSV structural proteins were further confirmed to be important in the activation of TLR4 as Walpita et al. demonstrated THP-1-differentiated macrophages induced the transcription of *IL6*, *TNF*, and *IL10* in a TLR4-dependent manner in response to RSV VLP, suggesting that VLPs could serve as effective vaccine platforms [153].

In regard to TLR2-RSV interaction, Murawski et al. demonstrated that while TLR4 and TLR2 in murine peritoneal macrophages both appear to have a role in the response to whole RSV, TLR2 plays the primary role in inflammatory cytokine production [154]. TNF, IL-6, CCL2, and CCL5 were all produced in response to RSV in a TLR4-dependent manner, but more robustly by TLR2 [154]. However, the type I IFN response was not TLR2-dependent [154]. This TLR2-dependent inflammatory response was also linked to the recruitment of neutrophils [154]. Furthermore, Murawski et al. demonstrated that mice deficient in TLR2 or TLR6 displayed significantly higher RSV burden, suggesting that TLR2/6 was critical in controlling RSV replication [154]. Segovia et al. further demonstrated that TLR2 and MyD88 in murine bone marrow-derived macrophages were not only involved in the production of inflammatory cytokines in response to RSV, but were also responsible for the first signal in inflammasome activation in response to RSV [15]. RSV activated production of pro-IL-1β and NLRP3 in mouse bone marrow-derived macrophages via the TLR2/MyD88/NF-κB pathways, whereas TLR4 did not appear to be involved (Figure 5) [15].

The role of TLR-mediated RSV recognition also remains under debate. Several studies demonstrated that TLR4 had no role in RSV clearance in HEK-293 cells [155] or responsiveness in primary human PBMCs [156], but IL-12 was critical for controlling RSV in C57BL/10 mice [157]. Triantafilou et al. demonstrated that only the knockdown of TLR4, not TLR2, reduced pro-IL-1β and IL-1β production in primary human lung epithelial cells [158]. This study did suggest that additional TLRs could be involved as the knockdown of TLR4 was not sufficient to abrogate the production of IL-1β [158]. While no direct link was established, the study suggested that the RSV short hydrophobic protein (viroporin) could be mediating this inflammasome activation in primary human lung epithelial cells [158]. These studies indicate that TLR-RSV recognition requires further investigation.

### 3.6. Baltimore Classification: Class VI

#### 3.6.1. Human Immunodeficiency Virus

Human immunodeficiency virus (HIV) causes acquired immunodeficiency syndrome (AIDS) and currently more than 38.4 million people worldwide are infected, with approximately 650,000 AIDS associated deaths in 2021 [159]. HIV consists of two-identical, single-stranded RNA genomes encoding 15 mature proteins [160]. The HIV envelope consists of the gp160 glycoprotein which is cleaved into gp120 and gp41 allowing HIV to bind CD4 and chemokine receptors for viral entry [161,162]. Decades of research have failed to develop an effective HIV vaccine. The phase III, RV144 clinical trial in Thailand demonstrated that the vaccine had ~60% efficacy within a year of vaccination, but this dwindled to 31% efficacy after 3.5 years [163].

Studies on HIV and surface TLR interactions are primarily association-based from HIV patient samples. Lester et al. identified that individuals with advanced disease expressed elevated levels of *TLR2* and *TLR4*, and increased *TLR6* expression was associated with increased plasma HIV RNA load [164]. Similar studies identified increased mRNA levels of *TLR2* and *TLR4* in hepatic tissue of HCV/HIV co-infected patients along with *TNF*, suggesting a role for innate immune recognition in the production of TNF [165]. Hernández et al. demonstrated that TLR2 and TLR4 were upregulated in HIV patients with opportunistic co-infections, but only TLR2 was upregulated in monocytes of HIV singly infected individuals suggesting that HIV co-infections may promote HIV replication via differential TLR expression [166]. Hernández et al. expanded on these observations and determined that cytokine expression was independent of HIV glycoproteins [167]. More recently, Vidyant et al. analyzed various single nucleotide polymorphisms in TLR2 and found that the TLR2 (-196 to -174 Ins/Del) polymorphism was at a higher prevalence in HIV infected individuals suggesting this polymorphism may be an HIV risk factor [168].

While Hernández et al. suggested that cytokine expression was independent of HIV glycoproteins, several studies demonstrate that HIV-related glycoproteins and other proteins induce an inflammatory response. Over several publications, Henrick et al. demonstrated how TLR2 and other surface TLRs interacted with various HIV related proteins including gp120, gp41, p17, and p24 [169,170,171]. Henrick et al. demonstrated that soluble TLR2 (sTLR2) was upregulated in human mammary epithelial cells and THP-1-differentiated macrophages in response to gp41, p17, and p24 and interacted with these proteins likely as a competitive inhibitor to reduce HIV-induced NF-κB activation [169]. This study further suggested that sTLR2 may play a role in reducing the transmission of HIV to infants during breastfeeding [169,172]. A subsequent study demonstrated that TLR2/1 was specifically involved in the recognition of p17 and gp41 and that TLR2/6 was responsible for the recognition of p24 in HEK-293 cells and HeLa cell-derived TZM-bl cells, while TLR4 was not involved in responding to any of these proteins (Figure 6) [170]. Exposure to HIV proteins differentially upregulated TLR2, TLR1, and TLR10 in human breast epithelial cell lines, THP-1-differentiated macrophages, and breast milk cells, but TLR10 was primarily responsible for the recognition of these proteins [171]. Henrick et al. confirmed that gp120 was not recognized by TLR2 and did not elicit a cytokine response [169,170,171]. In contrast to these studies demonstrating that HIV-related proteins activate an inflammatory response, Reuven et al. showed that gp41 inhibited the inflammatory TLR2/6-dependent response to lipoteichoic acid in both human THP-1 differentiated macrophages and murine RAW264.7 macrophages specifically through the GXXXG motif present in gp41 [173]. Additionally, in contrast to the Henrick et al. studies, Nazli et al. showed that gp120 induced a proinflammatory response including IL-8 and TNF production via TLR2 and TLR4 in primary human genital epithelial cells and required heparin sulfate [174]. While these studies identify several possible surface TLR ligands that activate an immune response, there remains controversy on the specific inflammatory response induced by each HIV-related protein.

#### 3.6.2. Mouse Mammary Tumor Virus

Mouse mammary tumor virus (MMTV) is an oncogenic retrovirus used to study human breast cancer [175]. MMTV is an enveloped, single-stranded, positive-sense RNA virus in the *Retroviridae* family. The envelope protein (Env) of MMTV contains a surface protein (SU) that binds the host receptor, transferrin receptor 1, and a transmembrane domain (TM) that is involved in membrane fusion [175]. MMTV is a particularly interesting virus due to its dependence on TLR4. Several studies have indicated that MMTV has a close relationship with the host gut microbiota and requires TLR4 and the host microbiota to persist from one mouse generation to the next [176,177]. MMTV can “steal” host CD14/MD-2/TLR4 and LPS binding protein (LBP) and maintain these proteins in its envelope [177]. This allows MMTV to associate with LPS resulting in the activation of host TLR4, production of IL-6, and subsequent activation of IL-10, which is thought to allow MMTV to evade the host immune response by activating an immunosuppressive environment in the host [176,177]. This is a particularly interesting example of how a virus modulates the host immune response via surface TLRs.

Other studies have suggested that MMTV-related proteins activate TLR4 apart from hijacking LPS. Rassa et al. identified that the activation of TLR4 in mice was independent of MMTV replication and that the MMTV Env protein SU domain was the activator of TLR4 (Figure 6) [178]. This group demonstrated that Env protein co-precipitated with TLR4, independent of virus attachment or fusion with the host cell [178]. Jude et al. demonstrated that both TLR4 and IL-10 were required for the maintenance of MMTV in mice but was not required for the initial infection or replication [179]. Importantly, they demonstrated that MMTV-induced activation of TLR4 and IL-10 production in murine dendritic cells and macrophages that promoted subversion of the host immune response and viral persistence [179]. Burzyn et al. and Courreges et al. further demonstrated a role for TLR4 in the maturation and migration of murine dendritic cells during MMTV infection [180,181]. As MMTV infects dendritic cells, this may serve as a mechanism of enhancing MMTV infection and spread [180,181].

### 3.7. Baltimore Classification: Class VII

#### Hepatitis B Virus

Hepatitis B virus (HBV) infections can occur early or late in life altering the outcome of viral infection [182]. Infections early in life typically result in lifelong chronic infections, while infection later in life can be self-resolving or, rarely, cause liver failure [182]. HBV is an enveloped, circular, partially double-stranded DNA virus in the *Hepadnaviridae* family. The HBV host-derived envelope contains several viral proteins including hepatitis B surface antigens (HBsAg), which are defined as large (L-HBsAg), middle (M-HBsAg), and small (S-HBsAg), surrounding the hepatitis B core antigen (HBcAg) [183]. The open reading frame (ORF) for HBcAg also encodes the pre-core region, namely hepatitis B e antigen (HBeAg), which is a secreted protein during infection [183,184,185]. HBsAg, HBeAg, and HBcAg can all be found in human serum during various stages of HBV infection, indicating that several virus-related proteins are present for the potential activation of surface TLRs [183,184,185,186]. Additionally, HBV secretes defective particles lacking genetic material primarily composed of HBsAg thought to act as a means of immune suppression [186].

The HBV literature contains much controversy as to the role of each HBV protein in immune activation or immune suppression. Jochum et al. published the first HBV TLR cell surface study in 1990 prior to the discovery of human TLRs. Early studies observed an inhibition of the LPS response by HBV proteins, including HBsAg [47,48,49,50], HBcAg [47], and HBeAg [187,188], suggesting that HBV proteins could interact with TLR4 primarily in human PBMCs and THP-1-differentiated macrophages. Cooper et al. identified that HBcAg induced a TLR2-dependent inflammatory response by altering gene expression of *TNF*, *IL6*, and *IL12B* in THP-1-differentiated macrophages [189]. HBcAg also induced NF-κB activation in HEK-293 cells expressing TLR2 and was enhanced when co-expressing TLR2/CD14. However, co-expression of TLR4/CD14 did not induce NF-κB activation [189]. Furthermore, it was shown that the arginine rich domain of HBcAg is required to induce TLR2-dependent inflammatory cytokine activation [189]. Li et al. demonstrated that TLR2-deficient mice displayed lower HBsAg, HBeAg, HBcAg, and HBV DNA in serum suggesting anti-HBV immunity was stronger in TLR2-deficient mice [190]. In Kupffer cells, IL-10 production was dependent on TLR2 and produced in response to HBcAg but not HBsAg [190]. Yi et al. also examined HBcAg in relation to human M1/M2 macrophage polarization [191]. HBcAg impaired M2 polarization in a TLR2-dependent manner by promoting the production of proinflammatory cytokines including TNF and IL-6 [191]. In contrast to Li et al., HBcAg did induce production of IL-10 and suggested that the inflammatory response induced by HBcAg could be favorable in the elimination of chronic hepatitis B virus by promoting a proinflammatory environment [191].

HBV proteins like HBsAg and HBeAg appear to also have immunosuppressive activity by inhibiting the inflammatory response due to canonically inflammatory ligands [47,48,49,50,187]. Visvanathan et al. demonstrated that chronic HBV patients displayed lower levels of TLR2 on Kupffer cells, hepatocytes, and peripheral monocytes during infection due to HBeAg [187]. Peripheral blood monocytes from these patients produced lower levels of TNF and IL-6 in response to Pam3Cys (TLR2/1 ligand) [187]. Furthermore, whole blood cells produced more TNF in response to HBeAg-negative HBV than the HBeAg-positive HBV, suggesting that immune suppression is mediated by HBeAg [187]. HBeAg, not HBcAg, inhibited p38 phosphorylation in response to both LPS and Pam3Csy, suggesting that HBeAg is able to interfere with both TLR4 and TLR2 activation during infection [187]. Lang et al. demonstrated that HBeAg, but not HBcAg, colocalized with TLR2, TRAM, and Mal in HEK-293 cells at the subcellular level and interfered with TLR2 and TLR4 signaling via TNF in response to Pam3Csy and LPS [188]. Specifically, they identified three amino acids conserved between HBeAg and the TIR domain that are necessary to disrupt the TIR:TIR binding needed for activation of TLR-related signaling [188]. Wang et al. demonstrated that THP-1-differentiated macrophages pretreated with HBsAg were unable to activate IL-12p70 in response to Pam3CSK4, but the production of IL-1β, IL-6, IL-8, and TNF were not (Figure 7) [50].

Unlike previous publications demonstrating an inhibitory effect of HBsAg, Song et al. demonstrated that primary monocytes from healthy donors induced the production of IL-1β, TNF, and IL-10 in response to HBsAg in a TLR2 and NF-κB-dependent manner [192]. Zhang et al. demonstrated that primary human hepatocytes exposed to infectious or UV-IA HBV produced IL-1β, TNF, and IL-6 [193]. The authors suggested that the inflammatory response could be mediated by HBsAg but did not test specific viral proteins [193]. Interestingly, a proinflammatory response has additionally been demonstrated in adaptive immune cells including B cells [194]. Li et al. used purified murine B cells to demonstrate that HBV induced upregulation of MHCII and CD86, increased secretion of IgM and IgG, and production of IL-6, IL-10, and TNF in a TLR2- and MyD88-dependent manner [194]. While there is debate as to whether HBV induces an inflammatory response or suppresses inflammation, contrasting results could be due to differences in the stage of infection analyzed, differences in infected cell types, and whether cells were exposed to a specific protein or infectious HBV.

## 4. Viruses at Toll Crossroads

Throughout the discovery of TLRs, there has been debate on TLR ligand specificity. One of the most prominent examples was whether TLR2 was involved in the recognition of LPS [195,196,197,198]. While studies clearly demonstrated that contaminating lipoprotein in LPS extractions was responsible for activating TLR2, research into whether TLR2 was involved in LPS recognition continued [199]. Similarly, there remains debate as to which surface TLRs are involved in the recognition of certain viruses. There is general agreement that TLR2 is involved in the recognition of several viruses including EBV [10,11,72,73,74], HIV [164,165,166,167,169,170,172,174], and PPV [43,44]. For HPV [53,54], MMTV [178,179,180,181], and EBOV [55,133,134,135,136,137], there is consensus that TLR4 is involved for recognition. TLR2, but not TLR4, is involved in recognition of CMV [13,14] and LCMV [140,141]. One factor that may reduce disagreement is the consistent use of specific cells. In the EBV studies, human cells, particularly HEK-293 cells, were used instead of murine cells to demonstrate an involvement for TLR2, which could explain the consistency seen between which TLR was identified as interacting with EBV or its proteins. Similarly, all the studies with MMTV used murine cells, primarily from C3H mice, to demonstrate the involvement of TLR4. In EBOV studies, primarily human cells including HEK-293 and THP-1 cells were used, but a few studies using C57BL/6 mice for infection models still demonstrated a role for TLR4.

TLR ligand specificity can remain conserved across species. Unlike studies using generally the same host cells, the two studies presented here examining PPV used cells from different hosts including HEK-293 cells and porcine PK-15 and both converged on an involvement for TLR2 [43,44]. This is interesting as it has been suggested there may be functional differences between human and porcine TLR2 [200]. TLR2 was also important for interaction with HPV in separate studies analyzing murine or human B cells. Again, this poses an interesting question in regard to ligand recognition across species as murine and human TLR2 are not identical and demonstrate differences in their ligand interaction [201,202]. For studies focusing on HIV, which mainly used primary human cells, there was agreement on a role for TLR2, but differences arose when defining the specific TLR2 heterodimer even when using the same types of cells. Overall, these studies suggest that consensus can be facilitated by using biologically relevant cells.

The viruses that stimulated the most disagreement regarding surface TLR interactions were RSV, HCV, HSV, DENV, and SARS-CoV-2. For RSV, it is difficult to determine why studies disagreed on the involvement of TLR2 or TLR4, and in some cases no TLR involvement at all. Some of these differences could be due to investigating a role for TLR4 in response to specific RSV proteins or infectious virus. Studies by both Segovia et al. and Triantafilou et al. utilized infectious RSV strain A2 and human cells, but Segovia et al. used HEK-293 cells and Triantafilou et al. used primary epithelial cells, which could explain the differences observed [15,158]. Studies with HCV primarily focused on *E. coli*-derived HCV proteins [17,19,21,22]. These studies demonstrated an involvement for TLR2, rather than TLR4 [17,19,21,22]. In addition, some of these studies listed non-detectable or low (<0.01 EU/mL) endotoxin [17,19]. The studies using HCV core protein produced in a peptide synthesizer showed no involvement for TLR2 or TLR4 [203], while HCV core protein produced in Raji cells demonstrated an involvement for TLR4 [52]. Therefore, investigation may be needed to determine how potential *E. coli*-derived contaminants like lipoproteins can affect TLR2 responses due to HCV core protein purification. In regard to DENV, Modhiran et al. demonstrated that recognition of DENV depended on TLR4, rather than TLR2/6 as had been previously demonstrated and attributed this difference to misfolded and/or contaminated DENV proteins [107,108]. Similarly, studies disagree on the involvement of TLR2 or TLR4 in the recognition of various SARS-CoV-2 ligands. In these studies, possible contamination from the different recombinant protein expression systems used including *E. coli*, insect cells, and HEK-293 cells, rather than different host/cell type, may be the major factor affecting identification of surface TLR SARS-CoV-2-derived ligands. Future studies should attempt to utilize highly purified viruses and ligands to prevent nonspecific TLR activation. Finally, studies will benefit by confirming results with different biologically relevant cell types since this will reduce variability introduced by differences in handling reagents and will facilitate cell-type specific comparisons between different groups.

## 5. Surface TLRs at the Forefront of Viral Pathogenesis

Toll-like receptors are indispensable for the recognition and response to pathogens, and while this recognition may intuitively benefit the host, it may also result in excess inflammation and host damage benefiting viral replication and survival. While there was disagreement as to whether surface TLR recognition of viruses benefits the host or the virus for RSV, HSV, HIV, and DENV, many of the viruses discussed here appear to benefit from surface TLR recognition. This negative impact on the host was clear for EBOV, in which TLR4 recognition was linked to severe inflammation, cell death, and viral dissemination. HBV also appears to possess several surface TLR inhibitory proteins to reduce host responsiveness. Studies also showed that surface TLR recognition benefited the host. Soluble TLR2 in breast milk protected infants from HIV-1 transmission. Studies also generally agree that TLR4 and potentially TLR2 are crucial for controlling RSV replication. However, the majority of studies reviewed here suggest that viral ligands are detrimental to the host response by either directly activating surface TLRs to cause detrimental inflammation or by inhibiting TLR activation to evade an immune response.

While this review focused on surface TLRs, viral nucleic acids are recognized by endosomal TLRs that may cooperate with surface TLR recognition of viruses during infection. Multiple nucleic acid sensing TLRs may be activated during viral replication in addition to surface TLRs during viral infection. For example, HSV is a double-stranded DNA virus with double-stranded RNA intermediates that has been shown to be recognized by nucleic acid sensing TLRs including TLR3 and TLR9 in addition to surface TLRs including TLR2 and TLR4 [204]. HSV studies discussed here demonstrate that TLR2 and TLR9 act in a compensatory manner or in combination, and together protect the host from HSV infection [26,30,33,37,38,91]. Similarly, TLR2 and TLR9 were involved in combination to respond to EBV and resulted in differential or additive regulation of cytokines [73,74]. Some viruses including the vaccine strain of YFV, YF-17D, activate several TLRs including TLR2, TLR7/8, and TLR9 resulting in an improved polyvalent immune response due to the activation of several TLRs [127]. The activation of multiple TLRs may serve to reduce the ability of viruses to subvert the host immune response by providing the host with several avenues for activating an anti-viral response.

## 6. Surface TLR Promiscuity or Sending Mixed Ligand Messages

Due to the vast diversity of ligands identified for TLR2 and TLR4, whether these surface TLRs are truly promiscuous and lack specificity remains controversial. While di- and tri-acylated lipoproteins have been considered the only “true” TLR2 ligands [205], the list of TLR2 and TLR4-associated ligands continues to expand. The contribution of contamination in surface TLR signaling arose with both TLR4 and TLR2, in which initially it was believed that LPS activated TLR2 as well as TLR4 [195,196]. However, several studies demonstrated that phenol extractable, trace levels of protein contaminants present in LPS preparations were responsible for the activation of TLR2 [197,198]. In regard to TLR2 specific ligands, several studies have attempted to address possible false positive ligand-induced TLR activation that may instead be a result of contamination due to the wide-spread use of *E. coli* derived proteins and VLPs [60,205,206,207,208]. As endotoxin is a primary concern regardless of the expression system used due to its prevalence in the environment [209,210,211], several methods, each with their own drawbacks, have been developed including endotoxin neutralization with polymyxin B [212,213]. In particular, ClearColi is used as an expression system for protein purification due to the lack of immunogenicity of its endotoxin in humans [214,215,216]. However, ClearColi endotoxin remains a murine TLR4 ligand [214,215,216]. Additionally, it is important to consider that neutralizing TLR4 or using TLR4-deficient cells or animal models does not necessarily remove the ability of the host to detect endotoxin, as caspase-11 in humans or caspase-4/5 in mice detects endotoxin [217]. Aside from endotoxin and lipoprotein contamination associated with *E. coli* expression systems, other expression systems present their own contaminants, such as residual baculovirus in insect cell systems [218], fertilizers and plant-specific sugars in transgenic plant systems [219], or animal virus and host-proteins in mammalian systems [219]. Therefore, contaminants from protein expression and purification systems remain a significant challenge in identifying specific TLR ligands. Several studies presented here utilized methods to address endotoxin, however, many of these studies did not check for contaminants other than endotoxin, particularly the presence of lipoprotein contamination (Appendix A). The ultra-purification of specific TLR ligands is not without challenge, but it is imperative to utilize ligands of high purity. Technologies like surface-plasmon resonance may facilitate identifying specific TLR ligands as integral membrane proteins become easier to study in artificial systems. Thus, it would benefit the field to proceed with caution when identifying new TLR ligands and address the role of contamination to avoid inadvertently inducing a response independent of the ligand of interest.

## 7. Conclusions

In this review, we endeavored to compile the body of literature focusing on the role of surface TLRs in viral recognition. We focused on studies demonstrating that surface TLRs directly interact with a specific virus or that specific viral proteins interfere with surface TLRs. In addition, we discussed potential differences between studies that could explain why the surface TLR responsible for viral recognition was not always consistent between studies. Finally, we addressed potential sources of contamination in protein and virus purification that can activate surface TLRs, which can confound results obtained from “purified” ligands. It is unlikely that all the results presented here are the result of contamination. Nonetheless, it is crucial that future studies confirm that TLR2 and TLR4 contaminating ligands commonly carried over from the expression systems used to purify ligands are not present to avoid misinterpretation of results. While significant research has been done in determining the role of surface TLRs in viral recognition, there is still substantial disagreement between studies and warrants further investigation into the role of surface TLRs in viral recognition. Although discovering specific viral ligands that activate host surface TLRs presents various challenges, determining if this initial recognition is key to a detrimental or beneficial outcome of infection is important to advance understanding of initial host-pathogen interactions. Such knowledge could lead to new avenues of therapeutic intervention.

## Figures and Tables

**Figure 1 viruses-15-00052-f001:**
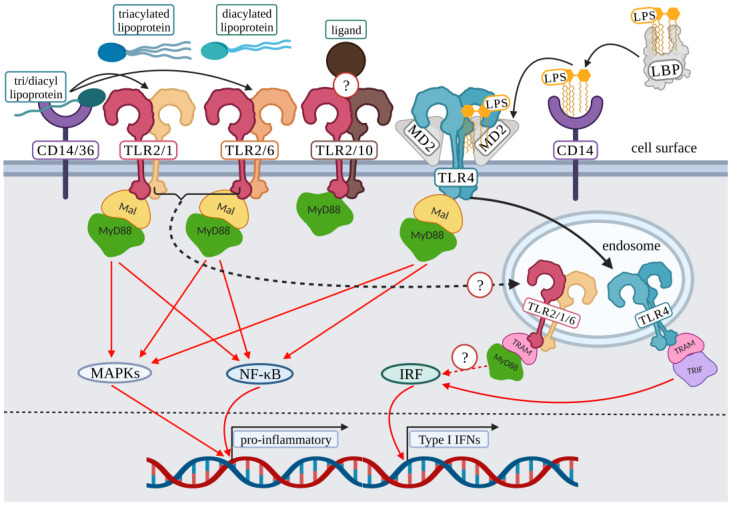
Toll-like Receptors (TLR) 2 and 4 Signaling Pathways: TLR2/1 recognizes triacylated lipoproteins. TLR2/6 recognizes diacylated lipoproteins. The ligand for TLR2/10 remains unknown. CD14 and CD36 cooperate in the transfer of lipoproteins to TLR2. TLR4 recognizes lipopolysaccharide (LPS) that is transferred to TLR4 by a series of proteins including LPS-binding protein (LBP), CD14, and myeloid differentiation factor 2 (MD-2). TLR2/1, TLR2/6, and TLR4 all signal via myeloid differentiation primary response 88 (MyD88) and Toll-interleukin-1 receptor (TIR) domain-containing adaptor protein (TIRAP/Mal) for the activation of a proinflammatory response through a mitogen-activated protein kinase (MAPK) or nuclear factor-κB (NF-κB)-dependent pathway. Endocytosis of TLR4 results in TLR4 signaling from the endosome via TRIF-related adaptor molecule (TRAM) and TIR-domain-containing adapter-inducing interferon-β (TRIF). TLR4 endosomal signaling activates type I interferons (IFNs) through interferon regulatory factors (IRF). TLR2 endosomal signaling has been suggested to occur via TRAM and MyD88 signaling for the induction of a type I IFN response. Signaling pathways poorly established in the literature are indicated by both dashed lines and question marks.

**Figure 2 viruses-15-00052-f002:**
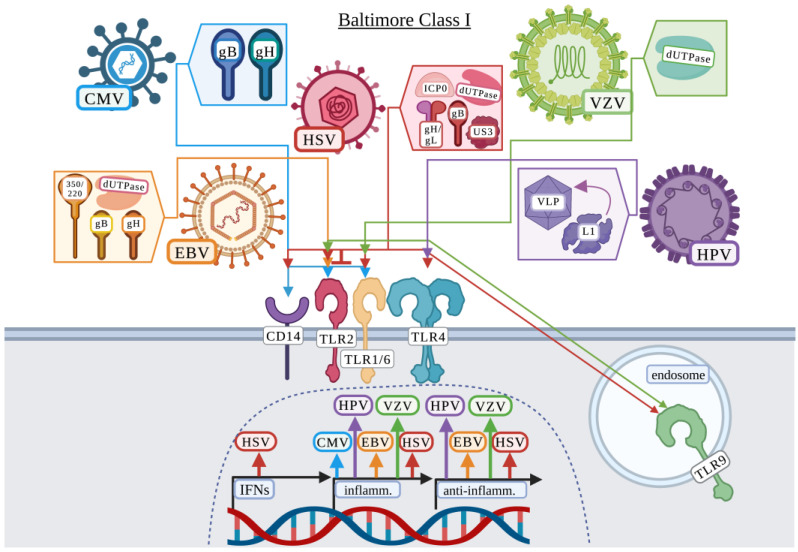
Baltimore Classification I viruses recognized by surface Toll-like Receptors (TLRs): within Baltimore Class I, 5 viruses were identified as activating a surface TLR response including cytomegalovirus (CMV), Epstein–Barr virus (EBV), herpes simplex viruses (HSV), human papillomavirus (HPV), and Varicella-zoster virus (VZV). CMV has been demonstrated to activate an inflammatory response via TLR2/1 in addition to CD14 and both gH and gB have been demonstrated to activate surface TLR signaling. EBV has been demonstrated to activate both an inflammatory and anti-inflammatory response via TLR2, with several potential ligands being identified including gp350/220, gB, gH, and dUTPase. HSV has several potential agonistic ligands including gH/gL, gB, dUTPase and antagonistic ligands including ICP0 and US3. These HSV ligands potentially interact, either agonistically or antagonistically with TLR2 with CD14, TLR1/6, TLR4, and TLR9 for an inflammatory, anti-inflammatory, and/or IFN response. HPV L1 composed VLPs induced an inflammatory and anti-inflammatory response via TLR4. VZV dUTPase activated a TLR2/1-dependent inflammatory and anti-inflammatory response, but VZV may have alternatively activated TLR9.

**Figure 3 viruses-15-00052-f003:**
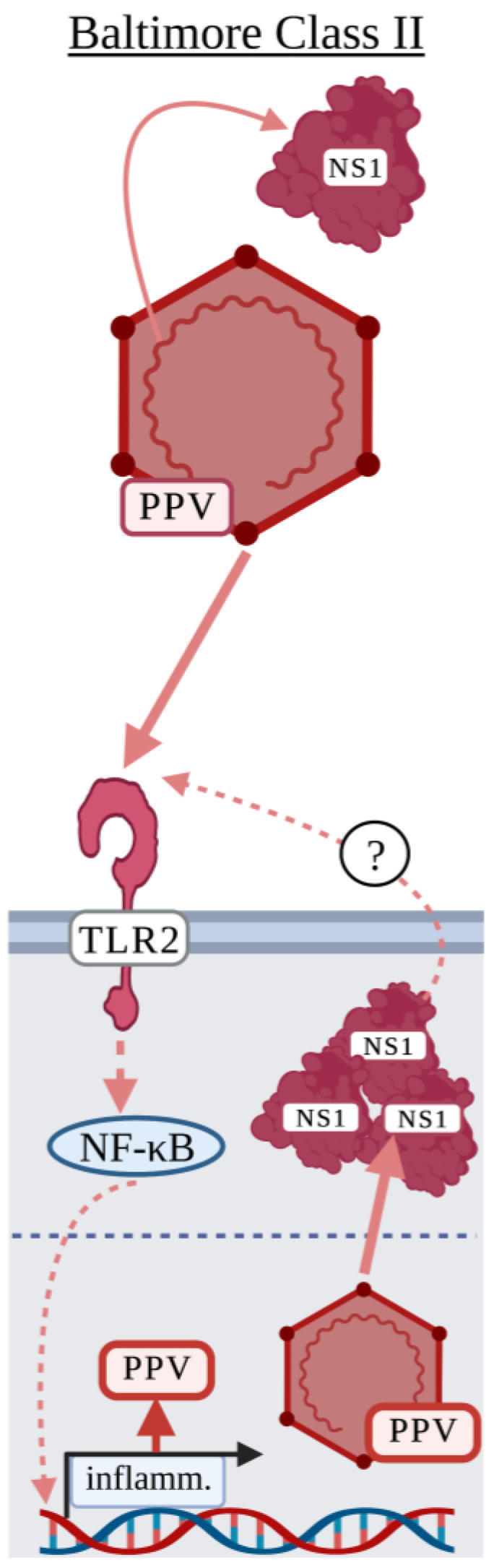
Baltimore Classification II viruses recognized by surface TLRs: for Baltimore Class II, Porcine Parvovirus (PPV) activated an inflammatory response via TLR2-NF-κB signaling, likely involving NS1.

**Figure 4 viruses-15-00052-f004:**
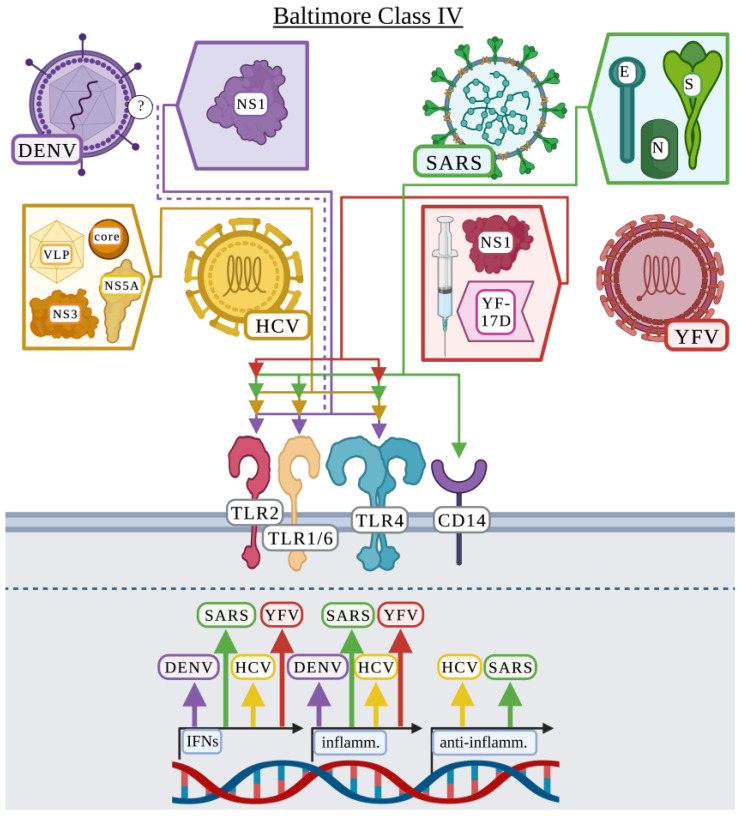
Baltimore Classification IV viruses recognized by surface TLRs: Baltimore Class IV include dengue viruses (DENV), hepatitis C virus (HCV), severe acute respiratory syndrome coronavirus (SARS), and yellow fever virus (YFV). All DENV serotypes were demonstrated to activate a surface TLR response. DENV was demonstrated to activate TLR2/6 and TLR4 for the induction of an inflammatory and IFN response, with NS1 being the potential ligand. HCV VLP, core, NS3, and NS5A were all demonstrated to activate different surface TLR responses including TLR2/1, TLR2/6, and TLR4 for an inflammatory, anti-inflammatory, and/or IFN response. SARS-CoV-2 ligands including S, E, and N were all demonstrated to activate an inflammatory, anti-inflammatory, and/or IFN response via TLR2/1 or TLR2/6 and TLR4. YFV NS1 activated an IFN response, and both the attenuated vaccine (indicated by the needle and syringe), YF-17D and NS1 activated an inflammatory response.

**Figure 5 viruses-15-00052-f005:**
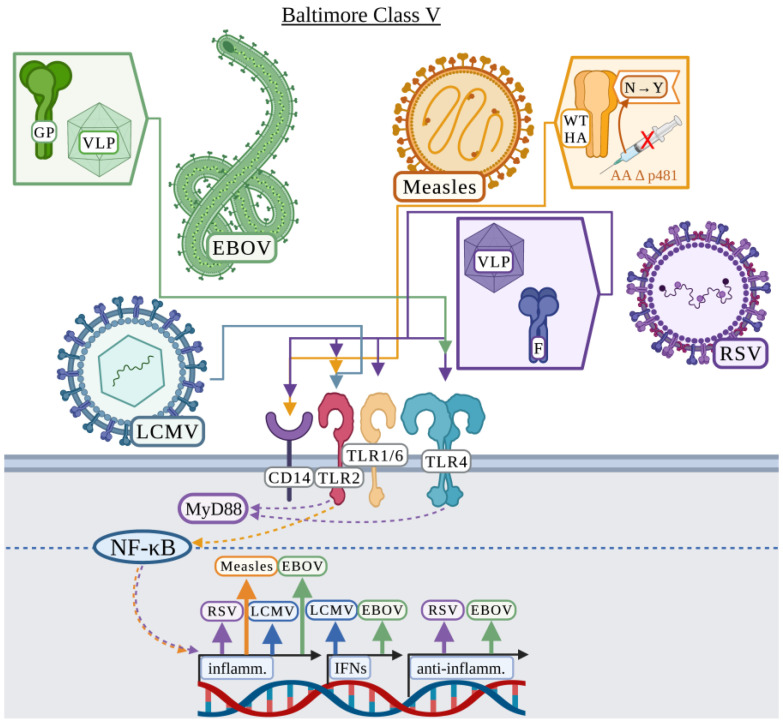
Baltimore Classification V viruses recognized by surface TLRs: the viruses in Baltimore Class V include Ebola virus (EBOV), lymphocytic choriomeningitis (LCMV), measles, and respiratory syncytial virus (RSV). The primary ligand identified for EBOV is GP, and VLPs composed of VP40, and GP have been demonstrated to activate an inflammatory, anti-inflammatory, and IFN response via TLR4. Specific ligands for LCMV have not been identified, but LCMV has been demonstrated to activate TLR2 for the induction of an inflammatory and IFN response. Wild-type measles HA activates TLR2 with CD14, while the HA of the measles vaccine strain (indicated by the needle and syringe) does not (indicated by an X), which is believed to be due to a single amino acid difference at position 481, an asparagine to tyrosine change. RSV F protein and VLP have been demonstrated to activate TLR2/6 along with CD14, and TLR4 via MyD88-NF-κB signaling for an inflammatory and anti-inflammatory response.

**Figure 6 viruses-15-00052-f006:**
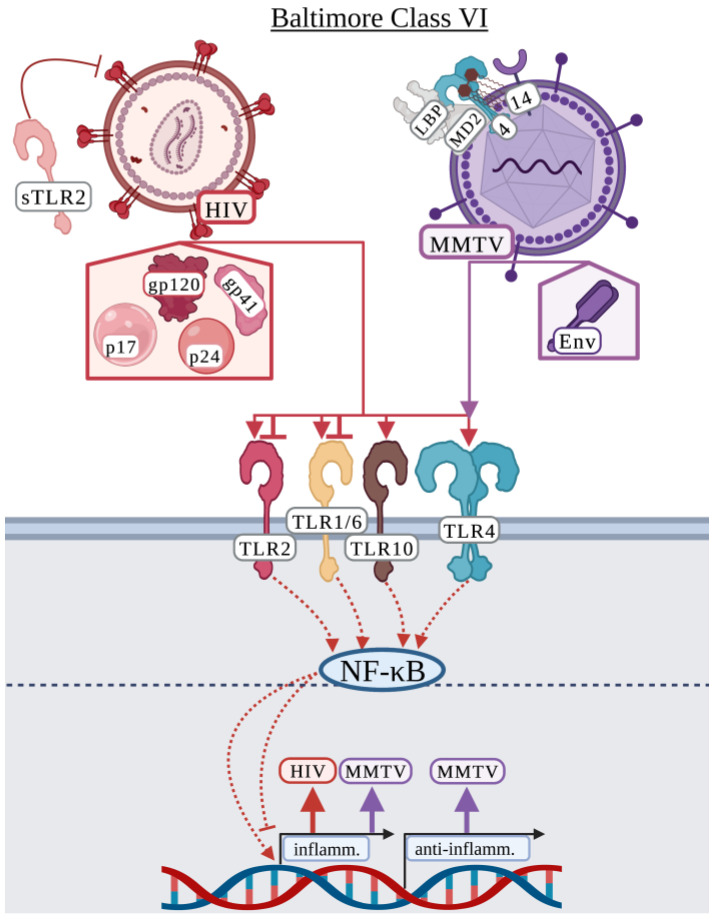
Baltimore Classification VI viruses recognized by surface TLRs: Baltimore Class VI include human immunodeficiency virus (HIV) and mouse mammary tumor virus (MMTV). HIV infection was shown to be blocked by soluble TLR2 (sTLR2) with HIV ligands being shown to act agonistically and antagonistically. HIV ligands including gp120, gp41, p17, and p24 either activated or inhibited TLR2/1, TLR2/6, and/or TLR4 for the activation or inhibition of an inflammatory response. MMTV is unique in that it ‘steals’ host associated TLR4, MD-2, LPS binding protein (LBP), and CD14 to associate with LPS to activate host TLR4 for both an inflammatory and anti-inflammatory response. In addition to this, MMTV has been suggested to activate TLR4 via its Env protein.

**Figure 7 viruses-15-00052-f007:**
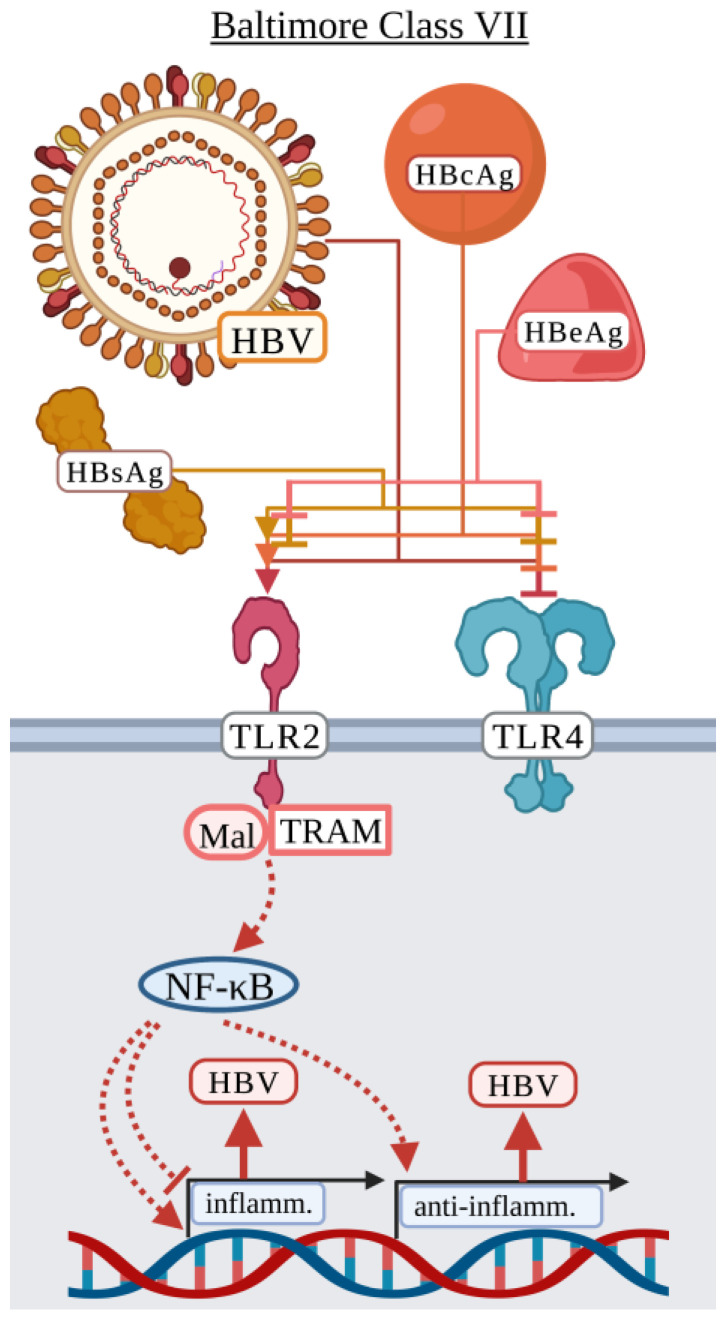
Baltimore Classification VII viruses recognized by surface TLRs: Baltimore Class VII includes hepatitis B virus (HBV). HBV ligands were agonistic and antagonistic including HBsAg, HBcAg, and HBeAg. HBsAg was found to both activate and inhibit TLR2 signaling and inhibit TLR4 signaling. HBcAg was found to activate TLR2 signaling and inhibit TLR4 signaling. HBeAg was shown to inhibit both TLR2 and TLR4 signaling by colocalizing with Mal and TRAM at the subcellular level.

## Data Availability

No new data were created or analyzed in this study. Data sharing is not applicable to this article.

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
