# Peer review of "Scratching the Surface Takes a Toll: Immune Recognition of Viral Proteins by Surface Toll-like Receptors"

_viruses, 2022, doi:10.3390/v15010052_

Round 1

Reviewer 1 Report

This manuscript gives an excellent review of the current literature on TLR recognition and signaling in response to viral infections. Only two minor comments below:

Section 2, lines 58 - 94, describes the signaling process of different TLRs. A diagram accompanying this paragraph showing these different pathways would be helpful for the reader.

The cell type that some studies were conducted in is not always clear. For example in line 672, it states "Exposure to HIV proteins upregulated TLR2, TLR1, and TLR10, but TLR10 was primarily responsible for the recognition of these proteins". There are a few examples of this throughout the manuscript that should be corrected.

Author Response

We appreciate the positive comments and feedback on our manuscript. Please refer below to the changes we have made based on your helpful suggestions.

"Section 2, lines 58 - 94, describes the signaling process of different TLRs. A diagram accompanying this paragraph showing these different pathways would be helpful for the reader."

  • We agree that it would be beneficial to add a figure to accompany that section so we have added a new figure (now Figure 1). This figure focuses on the TLRs discussed in the review and a brief overview of the potential signaling pathways. Please refer to the updated manuscript with track changes to see the figure in the manuscript along with the corresponding figure legend.

"The cell type that some studies were conducted in is not always clear."

  • We agree that there were examples throughout the manuscript where the cell type and species were not clear.
  • Regarding this sentence, "Exposure to HIV proteins upregulated TLR2, TLR1, and TLR10, but TLR10 was primarily responsible for the recognition of these proteins", we have now adjusted this to read "Exposure to HIV proteins differentially upregulated TLR2, TLR1, and TLR10 in human breast epithelial cell lines, THP-1 differentiated macrophages, and breast milk cells, but TLR10 was primarily responsible for the recognition of these proteins."
  • Additionally, we have added the cell type/species if one was not clearly listed throughout the manuscript. Please refer to the updated manuscript with track changes to see those cell type/species additions.

We appreciate your diligent review of our manuscript and the helpful feedback.

Reviewer 2 Report

The manuscript by Hatton and Guerra addresses the role of TLRs in recognition of viral proteins leading to immune defense. This manuscript is well organized and written however the draft has couple of minor points that need to be addressed:

  1. While the main goal of this review is to describe multiple viral ligands for activation of TLRs the authors could expand this by showing the impact on activation of immune cells and immune response. For example, SARS-CoV-2 ligands can induce TLR4-mediated cytokine storm production in macrophages and neutrophils leading to acute respiratory syndrome.
  2. In the conclusion part the authors could comment the biological relevance of activation of variety of TLRs by different viral ligands to clear specific viral infection.

Author Response

We appreciate your helpful comments on our manuscript, and the positive feedback. Please refer below to our changes based on your suggestions. 

"While the main goal of this review is to describe multiple viral ligands for activation of TLRs the authors could expand this by showing the impact on activation of immune cells and immune response. For example, SARS-CoV-2 ligands can induce TLR4-mediated cytokine storm production in macrophages and neutrophils leading to acute respiratory syndrome."

  • We agree that this would be a good addition to several viruses we discussed.
  • The following lines were expanded upon or added to the manuscript to highlight the impact of activation on immune cells and immune response.
    • Lines 233-237: "Importantly, this group previously demonstrated that the TLR2 single nucleotide polymorphism (SNP) R753Q was associated with a higher degree of HCMV replication and liver disease [68] and then demonstrated in vitro in HEK-293 cells that this TLR2 SNP impaired the activation of NF-kB and subsequent regulation of these cytokines in response to gB [14]." 
    • Lines 196-201: "Two additional studies further demonstrated that EBV induced a TLR2-dependent response resulting in production of inflammatory cytokines/chemokines including IL-8 and monocyte chemoattractant protein 1 (MCP-1) [73,74] and the anti-inflammatory cytokine, IL-10 [73] in primary human monocytes [73] and THP-1 differentiated macrophages [74] potentially contributing to tumorigenesis associated with EBV infection. 
    • Lines 608-610: "Yang et al. suggested that VLP-based HPV vaccines may provide the host with a T helper cell-independent humoral response benefitting CD4+-deficient patients that are prone to HPV disease [54]." 
    • Lines 708-710: "These studies suggested that PPV may utilize TLR2 to modulate the host response including inducing cytokine storm during PPV infection [43,44]."
    • Lines 828-830: "Both Chung et al. and Swaminathan et al. suggest that HCV core protein via TLR2 signaling may increase host susceptibility to microbial infection [20] and HIV infection [21]."
    • Lines 852-900: "Defective macrophage polarization in HCV-infected individuals may contribute to HCV persistence in patients [116]."
    • Specifically, regarding SARS-CoV-2, we included lines 978-983: "While a consensus has not been made on which SARS-CoV-2 proteins activate which surface TLRs, several studies suggest that SARS-CoV-2 proteins contribute to detrimental cytokine storm via TLR2 [40,125] or TLR4 [121,123] associated with severe disease during infection. In addition, studies have demonstrated that detrimental neuroinflammation and subsequent central nervous system pathology is associated with the activation of TLR2 [39], and/or TLR4 [39,122]."
    • Lines 1029-1032: "Several studies have indicated a role for TLR4 in the recognition of EBOV and that TLR4-EBOV interaction is detrimental to the host possibly by increasing vascular permeability and increased susceptibility to EBOV infection due to increased monocyte differentiation [55,133–137]."
    • Lines 1041-1045: "In terms of treatments against EBOV infection, Younan et al. demonstrated that eritoran, a TLR4 antagonist, could serve as a therapeutic strategy and increased mouse survival following EBOV infection by reducing viral burden [136]. In addition to therapeutics, Lai et al. suggested that GP may stimulate protective immunity through TLR4 activating an innate-adaptive immune response, which could benefit vaccine development [135]."
    • Lines 1207-1210: "Zhou et al. suggested that an exacerbated inflammatory response resulting in increased inflammatory cell recruitment to the brain may increase the risk of damage to the central nervous system during infection [141]."
    • Lines 1516-1519: "More recently, Vidyant et al. analyzed various single nucleotide polymorphisms in TLR2 and found that the TLR2 (-196 to −174 Ins/Del) polymorphism was at a higher prevalence in HIV infected individuals suggesting this polymorphism may be an HIV risk factor [168]."
    • Lines 1700-1703: "In contrast to Li et al., HBcAg did induce production of IL-10 and suggested that the inflammatory response induced by HBcAg could be favorable in the elimination of chronic hepatitis B virus by promoting a proinflammatory environment [191]."
    • Lines 1708-1709: "Peripheral blood monocytes from these patients produced lower levels of TNF and IL-6 in response to Pam3Cys (TLR2/1 ligand) [187]."
    • Lines 1709-1711: "Furthermore, whole blood cells produced more TNF in response to HBeAg-negative HBV than the HBeAg-positive HBV, suggesting that immune suppression is mediated by HBeAg [187]."
  • The following are examples that were present in the manuscript at submission we believe relate to the impact of TLR activation on immune activation. 
    • Lines 336-341: "The absence of TLR2 but not TLR4 increased the survival of mice after HSV-1 infection as none of the mice experienced paralysis or seizures [23]. Subsequently, Kurt-Jones et al. further suggested that TLR2 recognition of HSV-1 and HSV-2 is detrimental in neonates leading to increased production of IL-6 and IL-8 by neonatal blood cells, which could explain why sepsis syndrome is observed more frequently in neonates with HSV [24].
    • Lines 745-748: "Finally, Chen et al. demonstrated that TLR6-deficient mice displayed increased survival after DENV infection compared to wild-type mice, suggesting that TLR6 has a detrimental role in DENV pathogenesis [106]." 
    • Lines 750-753: "These studies demonstrated that NS1 played a role in vascular leakage and activating inflammatory cytokine expression and/or production including TNF and IL-6 but via TLR4 and not TLR2 in murine bone marrow-derived macrophage and human PBMCs [107,108]."
    • Lines 1012-1014: "They demonstrated that TLR2 played a role in the production of IL-12p40 and IL-6 and that TLR2 may regulate the Th1/Th2 balance in response to YF-17D in mice and murine and human dendritic cells [127]."
    • Lines 1200-1201: "Zhou et al. demonstrated that TLR2-deficient mice did not have altered viral clearance but were unable to produce MCP-1 and IFN-a [140]."
    • Lines 1354-1355: "TLR4 recognition was critical for the control of RSV infection [51]."
    • Lines 1357-1360: "Several studies also established a role for TLR4 during RSV infection, including Tulic et al. that demonstrated that the TLR4 mutations Asp299Gly and Thr399Ile were associated with severe RSV bronchitis in infants [148–151]."
    • Lines 1511-1514: "Hernández et al. demonstrated that TLR2 and TLR4 were upregulated in HIV patients with opportunistic co-infections, but only TLR2 was upregulated in monocytes of HIV singly-infected individuals suggesting that HIV co-infections may promote HIV replication via differential TLR expression [166]."
    • Lines 1527-1529: "This study further suggested that sTLR2 may play a role in reducing the transmission of HIV to infants during breastfeeding [169,172]."
    • Lines 1557-1596: "This allows MMTV to associate with LPS resulting in the activation of host TLR4, production of IL-6, and subsequent activation of IL-10, which is thought to allow MMTV to evade the host immune response by activating an immunosuppressive environment in the host [176,177]."
    • Lines 1604-1606: "Importantly, they demonstrated that MMTV-induced activation of TLR4 and IL-10 production in murine dendritic cells and macrophages that promoted subversion of the host immune response and viral persistence [179]."
    • Lines 1608-1610: "As MMTV infects dendritic cells, this may serve as a mechanism of enhancing MMTV infection and spread [180,181]."
    • Lines 1694-1696: "Li et al. demonstrated that TLR2-deficient mice displayed lower HBsAg, HBeAg, HBcAg, and HBV DNA in serum suggesting anti-HBV immunity was stronger in TLR2-deficient mice [190]."

"In the conclusion part the authors could comment the biological relevance of activation of variety of TLRs by different viral ligands to clear specific viral infection."

  • We agree that this would be a beneficial addition to the manuscript's conclusion. The following paragraph is now included in the updated manuscript from lines 1985-1999:
    • While this review focused on surface TLRs, viral nucleic acids are recognized by endosomal TLRs that may cooperate with surface TLR recognition of viruses during infection. Multiple nucleic acid sensing TLRs may be activated during viral replication in addition to surface TLRs during viral infection. For example, HSV is a double-stranded DNA virus with double-stranded RNA intermediates that has been shown to be recognized by nucleic acid sensing TLRs including TLR3 and TLR9 in addition to surface TLRs including TLR2 and TLR4 [204]. HSV studies discussed here demonstrate that TLR2 and TLR9 act in a compensatory manner or in combination, and together protect the host from HSV infection [26,30,33,37,38,91]. Similarly, TLR2 and TLR9 were involved in combination to respond to EBV and resulted in differential or additive regulation of cytokines [73,74]. Some viruses including the vaccine strain of YFV, YF-17D, activate several TLRs including TLR2, TLR7/8, and TLR9 resulting in an improved polyvalent immune response due to the activation of several TLRs [127]. The activation of multiple TLRs may serve to reduce the ability of viruses to subvert the host immune response by providing the host with several avenues for activating an anti-viral response.

We are grateful for your diligent review of our manuscript.